# Comprehensive Genomic Profiling Reveals Diverse but Actionable Molecular Portfolios across Hematologic Malignancies: Implications for Next Generation Clinical Trials

**DOI:** 10.3390/cancers11010011

**Published:** 2018-12-21

**Authors:** Natalie Galanina, Rafael Bejar, Michael Choi, Aaron Goodman, Matthew Wieduwilt, Carolyn Mulroney, Lisa Kim, Huwate Yeerna, Pablo Tamayo, Jo-Anne Vergilio, Tariq I. Mughal, Vincent Miller, Catriona Jamieson, Razelle Kurzrock

**Affiliations:** 1Department of Medicine, Division of Hematology/Oncology and Center for Personalized Cancer Therapy, University of California San Diego, 3855 Health Science Drive #0987, La Jolla, CA 92093, USA; rabejar@ucsd.edu (R.B.); mychoi@ucsd.edu (M.C.); a1goodman@ucsd.edu (A.G.); mwieduwilt@ucsd.edu (M.W.); camulroney@ucsd.edu (C.M.); lit003@ucsd.edu (L.K.); cjamieson@ucsd.edu (C.J.); rkurzrock@ucsd.edu (R.K.); 2Department of Medicine, Division of Blood and Marrow Transplantation, University of California San Diego, La Jolla, CA 92093, USA; 3Department of Medicine, Division of Statistical Physics, University of California San Diego, La Jolla, CA 92093, USA; hyeerna@ucsd.edu (H.Y.); ptamayo@ucsd.edu (P.T.); 4Foundation Medicine Inc., Cambridge, MA 02141, USA; jvergilio@foundationmedicine.com (J.-A.V.); tmughal@foundationmedicine.com (T.I.M.); vmiller@foundationmedicine.com (V.M.); 5Tufts University Medical Center, Boston, MA 02111, USA

**Keywords:** next generation sequencing, lymphoid malignancies, myeloid malignancies, precision medicine

## Abstract

*Background*: The translation of genomic discoveries to the clinic is the cornerstone of precision medicine. However, incorporating next generation sequencing (NGS) of hematologic malignancies into clinical management remains limited. *Methods*: We describe 235 patients who underwent integrated NGS profiling (406 genes) and analyze the alterations and their potential actionability. *Results*: Overall, 227 patients (96.5%) had adequate tissue. Most common diagnoses included myelodysplastic syndrome (22.9%), chronic lymphocytic leukemia (17.2%), non-Hodgkin lymphoma (13.2%), acute myeloid leukemia (11%), myeloproliferative neoplasm (9.2%), acute lymphoblastic leukemia (8.8%), and multiple myeloma (7.5%). Most patients (*N* = 197/227 (87%)) harbored ≥1 genomic alteration(s); 170/227 (75%), ≥1 potentially actionable alteration(s) targetable by an FDA-approved (mostly off-label) or an investigational agent. Altogether, 546 distinct alterations were seen, most commonly involving *TP53* (10.8%), *TET2* (4.6%), and *DNMT3A* (4.2%). The median tumor mutational burden (TMB) was low (1.7 alterations/megabase); 12% of patients had intermediate or high TMB (higher TMB correlates with favorable response to anti-PD1/PDL1 inhibition in solid tumors). In conclusion, 96.5% of patients with hematologic malignancies have adequate tissue for comprehensive genomic profiling. Most patients had unique molecular signatures, and 75% had alterations that may be pharmacologically tractable with gene- or immune-targeted agents.

## 1. Introduction

Recent advances in next generation sequencing (NGS) have allowed for unprecedented insights into the genomic alterations that underlie oncogenesis, tumor biology, and survival. NGS permits identification of genomic alterations that not only inform a more granular subclassification of disease with various prognostic and predictive features but may also guide therapy selection [1]. Cumulative gains in understanding of cancer genomics and immuno-oncology are being rapidly translated into clinical practice, particularly for metastatic solid tumors, shifting the treatment paradigm from cytotoxic chemotherapy to a biologically informed approach where oncogenic alterations are matched with targeted agents [2,3]. The drug-target pairing that aligns a deranged molecular pathway with a cognate therapeutic agent constitutes the hallmark of precision medicine, and has demonstrated superior response rates as compared to nonselective chemotherapy in tumors such as melanoma, lung cancer, and chronic myelogenous leukemia [4,5]. There is now a plethora of genotype-matched therapeutics that have been Food and Drug Administration (FDA)-approved for the treatment of advanced solid tumors across a wide array of histologies.

Matched targeted therapy has also proven effective in several hematologic malignancies. For instance, chronic myelogenous leukemia (CML) is the poster child for a disease transformed by matched targeted therapy. Indeed, imatinib mesylate, which inhibits the enzymatic activity of the aberrant *BCR-ABL1* kinase, the hallmark of CML, has extended overall survival to near-normal life expectancy. Despite the unequivocal success of tyrosine kinase inhibitors (TKI) in CML, these types of practice-changing, rationally developed, targeted agents have not been broadly implemented in the treatment of many lymphoid and myeloid malignancies. As a result, despite the vast heterogeneity of hematopoietic tumors, most of the patients suffering from these disorders are still treated with non-specific cytotoxic chemotherapy. While this “one size fits all” approach is effective for many patients, it is clear that a significant proportion of patients, particularly those with adverse prognostic features, often relapse. The poorer outcomes are arguably because, despite cytotoxic chemotherapy, the driver oncogenic alterations persist, uninhibited, in the residual post-chemotherapy clones. In the relapsed setting, the standard salvage regimens rely on more intensive chemotherapy, followed by hematopoietic stem cell transplantation (HSCT), often associated with poor tolerability and significant morbidity, particularly in older patients [6]. Thus, understanding the molecular crosstalk that leads to disease progression and selectively targeting alterations within these signaling pathways remains a critical and unmet medical need.

To date, intense investigation into molecular underpinnings of hematologic disease has led to the identification of key recurrent mutations such as *MYD88*^L265P^ in lymphoplasmacytic lymphoma (LPL), *BRA*F^V600E^ in hairy cell leukemia (HCL), and *FLT3* in acute myelogenous leukemia (AML) that can be successfully targeted with cognate agents—ibrutinib, vemurafenib, and midostaurin respectively; producing encouraging results [7,8,9]. Undoubtedly, further insight into the genomic landscape of hematologic malignancies may help elucidate key molecular alterations and potentially inform treatment selection.

Herein, we analyze the genomic profiles of 235 patients with various hematologic malignancies for whom comprehensive genomic profiling (CGP) had been performed. We examined the mutational burden as well as the type and frequency of potentially actionable alterations across a wide variety of hematologic diagnoses to better delineate the mutational landscape of these disorders, providing a foundation for precision medicine trials in the hematologic malignancy clinical setting.

## 2. Results

### 2.1. Patient Characteristics

Tumor samples from 235 patients (133 men (59%) and 94 women (41%)) were collected. Of these, 227 subjects (96.5%) had adequate tissue quantity and purity for genomic analysis. Thirty patients had no genomic alterations, while 87% of patients (197/227) demonstrated at least one molecular alteration (variants of unknown significance (VUSs) were excluded) (Figure 1).

The most common malignancy in this cohort was myelodysplastic syndrome (22.9%), chronic lymphocytic leukemia (17.2%), non-Hodgkin lymphoma (13.2%), acute myeloid leukemia (11%), myeloproliferative neoplasm (9.2%), acute lymphoblastic leukemia (8.8%), and multiple myeloma (7.5%) (Table 1). The most common tissue source for NGS analysis was peripheral blood (*N* = 99 patients (44%)) or bone marrow (*N* = 86 (38%)), followed by lymph node (*N* = 17 (7%)) and other (*N* = 25 (11%)).

### 2.2. CGP Results

The genomic landscape of distinct alterations identified across hematopoietic malignancies is depicted in Figure 2.

The median number of alterations detected per patient was 3 (range, 0–14). The majority of patients (*N* = 197/227 (87%)) harbored at least one alteration. Of the total distinct aberrations, alterations in *TP53* (*N* = 59), *TET2* (*N* = 25) and *DNMT3A* (*N* = 23) were among the most frequent aberrations. (Figure 3 and Figure 4).

A total of 546 distinct alterations were identified by NGS in the entire cohort of 227 patients (Table 1). Types of alterations identified included substitutions, indels, copy number alterations (CNAs), and gene fusions. The most frequent types of alterations were mutations (85% of all alterations (594/698)), followed by fusion/rearrangement (7% (48/698)), copy loss (6% (42/698)), and copy gain/duplication/amplifications (2% (14/698)) (Figure 5).

In at least one patient, one hundred and forty-eight distinct genes were altered. The vast majority of patients (95.5%) had distinct molecular profiles, 4.5% of patients had identical molecular signatures (solitary alterations that were identical in at least one other patient involving the following genes: *JAK2* V617F (.5%), *TP53* R273H (1%), *SF3B1* K700E (1%)). (An identical molecular portfolio/signature implies that both the genes involved and the precise loci altered in those genes were identical).

### 2.3. TMB

In our cohort, 219 patients had data for TMB. TMB ranged from 0.4 to 140, with a median TMB of 1.7 mutations per megabase. The majority of patients (84%) had low TMB (≤1–5 mutations/MB); 12%, intermediate TMB (>5 to ≤19 mutations/MB); and only 2%, high TMB (≥20 mutations/MB) (Figure 6). TMB of myeloid neoplasms was lower than that for lymphoid malignancies (median = 0.9 mutations/MB (range, 0.8–12) versus 2.5 (range, 0.4–140) (*p* = 0.0012).

### 2.4. Actionable Alterations

Potentially actionable mutations were identified in 75% of patients with adequate tissue (170/227). Of these, 32 patients (14% of the 227) had ≥1 alteration theoretically actionable by an on-label FDA approved drug; 112 patients (49% of 227), by an off-label FDA-approved drug; and, of the remainder, 26 patients had an alteration actionable by an experimental drug. Pharmacologically tractable aberrations were found across all hematological malignancies (Table 2). 

## 3. Discussion

Our study demonstrates that the majority of hematologic tumors (~97%) had adequate tissue for CGP interrogation. Further, of the 227 patients with adequate tissue, 75% harbored alterations that could be prosecuted by a drug already in the clinical setting (Figure 1) (by an FDA-approved, on-label drug (14%) or by an FDA-approved off-label drug (an additional 49% of patients) with the rest targetable by experimental drugs in clinical trials). This finding is similar to observations in solid tumors where up to 70% of patients have an alteration that was theoretically pharmacologically tractable with an approved drug, and over 90% of patients have a potentially druggable alteration if experimental drugs are included [39]. In contrast to solid tumors, however, with few exceptions, the clinical utility of CGP for therapeutic decision-making for hematological malignancies has been limited and genomic information has largely been confined to diagnosis, classification, and prognostication. An example of such use is *TP53* or *ATM* mutations in chronic lymphocytic leukemia (CLL), which predict resistance to or short duration of response to chemotherapeutic agents [40]. Still, therapeutic matching in AML is beginning to be reported, with notable examples being the FLT3 inhibitor midostaurin and gilteritinib or the isocitrate dehydrogenase (IDH) inhibitors for patients with cognate aberrations [9,27]. Overall, however, master precision medicine studies matching patients to diverse cognate agents based on CGP are now being widely performed across solid tumors [2,3] but are in nascent stages in the hematologic field.

Interestingly, as in solid tumors, there was great diversity in genomic portfolios in our patients [1]. Indeed, 148 distinct genes were altered in at least one patient and there were 546 unique genomic alterations (Table 1). The vast majority of patients (95.5% of the 227 patients with adequate tissue) had distinct genomic signatures. These findings suggest that customized combinations of drugs may be needed for optimal matching.

The successful application of molecularly informed therapy has been demonstrated in Waldenström’s macroglobulinemia (WM) lymphoplasmacytic lymphoma (LPL) where about 90% of patients have a dominant mutation in *MYD88*^L265P^, an adapter protein used by toll-like receptors that mediate signaling through Bruton tyrosine kinase (BTK) to promote proliferation and survival [41]. In a phase 2 trial, inhibition of BTK with ibrutinib achieved an overall response rate of 91% in previously treated patients with WM [7]. Furthermore, as expected, the response rate to ibrutinib was significantly higher in patients with *MYD88* mutations vs. wild-type *MYD88* genotype [7]. In our cohort, the *MYD88*
^L265P^ mutation was found in one patient with WM as well as in four (of 18) patients with diffuse large B-cell lymphoma (DLBCL), including one patient with a primary central nervous system lymphoma (PCSNL). In the literature, *MYD88* mutations have also been identified in marginal zone lymphoma (MZL), CLL, DLBCL and PCNSL [42,43,44,45]. Preliminary data from a phase I trial of single-agent ibrutnib in four patients with PCNSL demonstrated responses in two of the three patients evaluated [46]. Our data further confirms that mutations in *MYD88* are readily identified by NGS. Trials in patients with diverse malignancies and *MYD88* mutations with BTK inhibitors may be warranted.

Activating mutations in *BRAF* were found in CLL (*N* = 3 of 39 patients; two with *BRAF* G469A and one with *BRAF* V600E), multiple myeloma (*N* = 1 of 17; *BRAF* V600E), hairy cell leukemia (HCL) (*N* = 1 of 1; *BRAF* V600E) and Erdheim Chester disease (ECD) (*N* = 1 of 1 patient; *BRAF* V600E). These mutations are potentially targetable by the BRAF inhibitors vemurafenib and dabrafenib and the MEK inhibitors trametinib and cobimetinib [4,17]. *BRAF* alterations have been identified as a dominant driver mutation and as a biomarker for sensitivity to BRAF inhibition in HCL, ECD, and myeloma [8,47,48,49]. BRAF alterations have been noted previously in 3% of patients with CLL [50]. In our studies, two CLL patients (5%) had mutations leading to alanine to glycine substitution at position 469, which is an activating mutation in other tumors and confers sensitivity to BRAF inhibition [51]. To our knowledge, this mutation has not been targeted in CLL previously. More recently, Wander et al. also reported a t-AML patient with a *BRAF* V600E mutation, who was refractory to several induction regimens, yet demonstrated a remarkable response to combined targeted BRAF/MEK therapy (with dabrafenib and trametinib) as evidenced by restoration of normal hematopoiesis, clearance of peripheral blasts, and a significant reduction in marrow leukemic burden with a concordant decrease in the *BRAF* V600E allelic burden [52]. Although transient, this patient’s response serves as a proof of concept that targeting oncogenic driver mutations may have similar efficacy in hematologic malignancies as in solid tumors supporting the emerging paradigm of designing targeted therapies based on the presence of actionable lesions [53].

TMB has been shown to correlate to response with checkpoint inhibitors in solid tumors [54]. The vast majority of our 219 patients in whom TMB could be evaluated had a low TMB, but 12% (*N* = 26) had an intermediate burden and 2% (*N* = 5 patients) had high TMB. Further, the mutational burden was significantly higher in lymphoid versus myeloid malignancies (2.5 versus 0.9 mutations/MB, respectively (*p* = 0.0012). This data is consistent with our previous analysis that showed that 32% of patients with lymphoid malignancies had intermediate to high TMB [55]. Three of our patients with higher TMB had mismatch repair gene alterations; in the solid tumor field, the PD-1 inhibitor pembrolizumab has recently received approval for all patients with solid tumors and microsatellite instability-high disease (which is due to mismatch repair alterations and is associated with a high TMB and high response rate to anti-PD1 agents [56]. These results suggest that there may be a subset of patients with hematologic malignancies who are amenable to response to checkpoint inhibitors as well.

Despite a wealth of clinical experience with molecularly matched therapies in solid tumor oncology, some of these targetable alterations remain unexplored within the clinical realm of hematology and present one of the major limitations of our study. This is partly due to the fact that many of the theoretically applicable molecules are FDA-approved primarily in solid tumors [57]. Furthermore, there is still an incomplete consensus regarding what represents a targetable alteration, and the necessary level of evidence needed to support the use of cognate agents in the clinic remains a matter of debate. Thus, further studies assessing the role of therapy matched to genomic portfolios across hematologic malignancies are certainly warranted.

## 4. Patients and Methods

### 4.1. Patients

We analyzed the genomic alterations by CGP and clinical characteristics of 235 patients with diverse hematologic cancers seen at the UCSD Moores Cancer Center (La Jolla, CA, USA) from October 2012 through December 2016. This study was performed and consents obtained in accordance with the guidelines of the UCSD Institutional Review Board (NCT02478931) (Center for Personalized Cancer Therapy) (Center for Personalized Cancer Therapy).

### 4.2. Comprehensive Genomic Profiling (CGP)

We conducted CGP on tumor samples from lymph nodes, peripheral blood, bone marrow, or tissue using FoundationOne Heme^®^ (F1H; Foundation Medicine Inc., Cambridge, MA, USA), a clinical grade, high-throughput, hybridization capture-based NGS assay for targeted sequencing of all exons of 406 genes as well as RNA sequencing of 265 genes. F1H is a validated clinical laboratory improvement amendments (CLIA)-approved, NY-state approved assay, and the methods used have been previously reported in detail [58,59]. It is capable of simultaneously identifying all genomic alterations, including insertions/deletions, base pair substitutions, copy number alterations (CNA), and select gene rearrangements. Variants of unknown significance (VUSs) were not included in any of our analyses except for tumor mutational burden (TMB) assessment.

### 4.3. Tumor Mutational Burden (TMB)

For TMB, the number of somatic mutations detected on NGS (interrogating 1.2 megabase (Mb) of the genome) are quantified and that value extrapolated to the whole exome using a validated algorithm [60,61]. Alterations likely or known to be bona fide oncogenic drivers and germline polymorphisms are excluded. TMB was measured in mutations per MB. TMB levels were divided into three groups (per Foundation Medicine template): Low (1–5 mutations/MB), intermediate (6–19 mutations/MB), and high (≥20 mutations/MB).

### 4.4. Definition of a Potentially Actionable Alteration

An alteration was designated as potentially actionable if there is ≥1 FDA-approved drug(s) or experimental compounds in a clinical protocol that may impact the function of the protein product of the alteration or its immediate downstream effectors, or that differentially distinguishes the protein in cancerous versus normal cells. Small molecule inhibitors with 50% low inhibitory concentration for the target and antibodies that recognize the protein were considered as impacting the target.

### 4.5. Data Analysis and Statistics

Pertinent data including patient demographics, tumor histology, and tissue source as well as molecular testing results, number, and type of genomic alterations, were extracted from patients’ electronic medical records. Descriptive statistics (medians, means, ranges, and frequencies) were used.

## 5. Conclusions

In conclusion, we found that most patients with hematologic malignancies exhibit complex molecular profiles that capture an array of oncogenic pathways across various histologies. The vast majority of individuals have at least one or more genomic alterations that are potentially actionable with existing drugs. A small subset of patients have intermediate/high tumor mutational burden, a variable that has previously correlated with response to checkpoint inhibitor immunotherapy. These observations present a window of opportunity for clinical trials to rationally test the application of genomically targeted therapeutics or immunotherapy, particularly in relapsed/refractory patients who have either exhausted or are unable to tolerate standard chemotherapy.

## Figures and Tables

**Figure 1 cancers-11-00011-f001:**
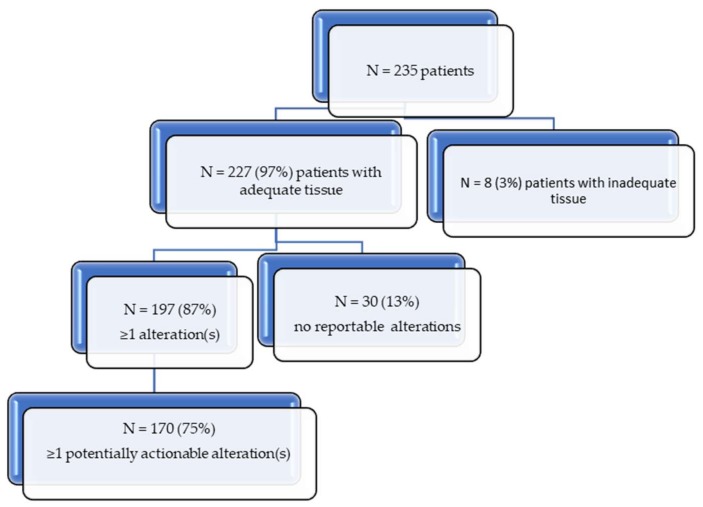
Consolidated Standards of Reporting Trials (CONSORT) diagram.

**Figure 2 cancers-11-00011-f002:**
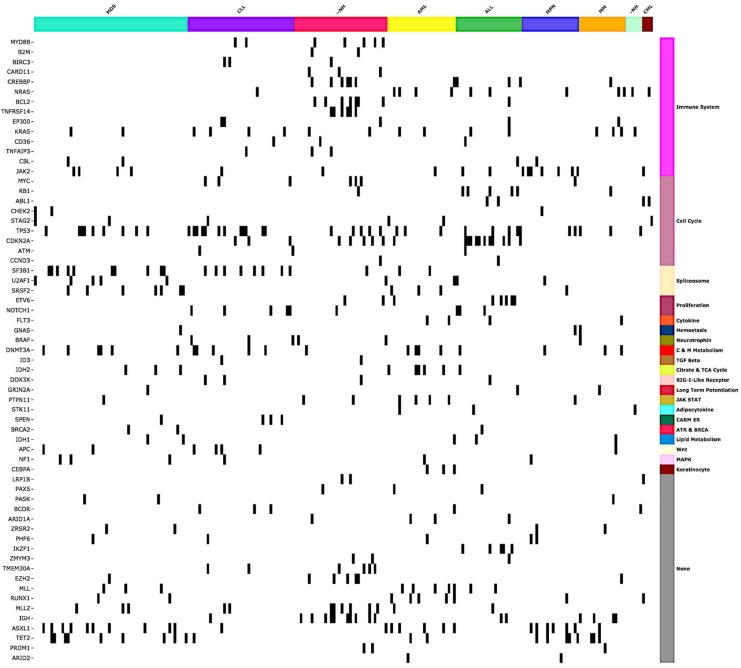
The genomic landscape of distinct, clinically relevant gene alterations across hematologic cancers. Molecular alterations are organized by gene sets derived from MSigDB Collection2 (Version 6.1) [10,11,12].

**Figure 3 cancers-11-00011-f003:**
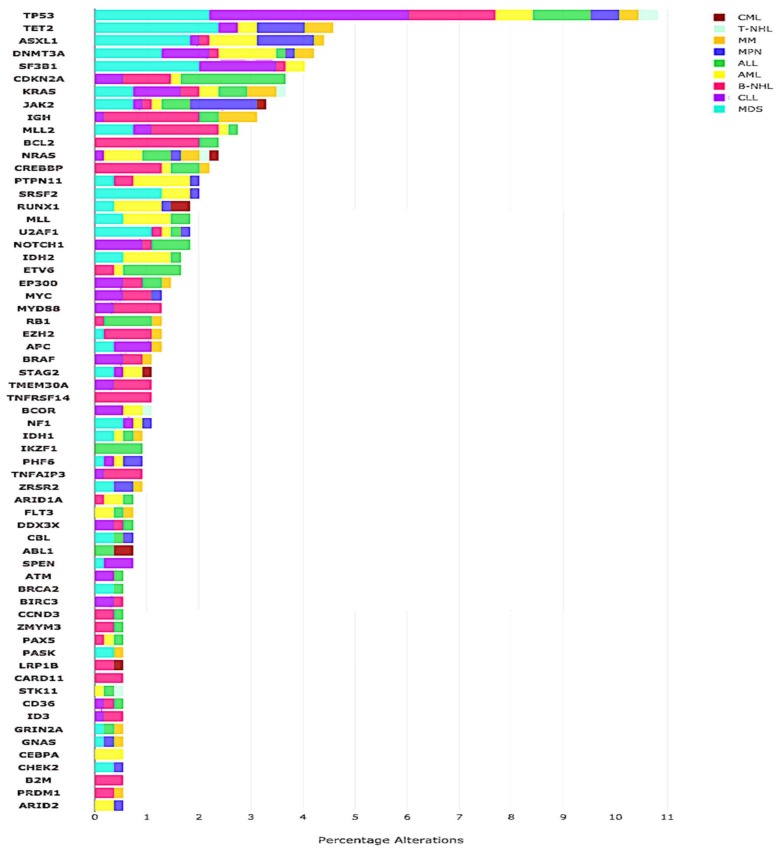
Frequency of most common alterations across various hematologic histologies.

**Figure 4 cancers-11-00011-f004:**
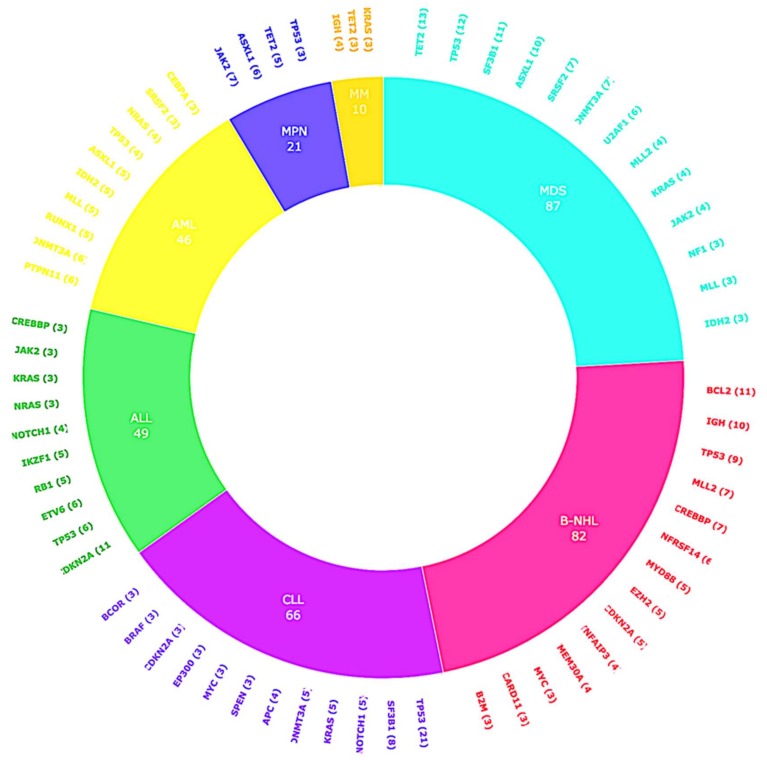
Overview of hematopoietic malignancies and most frequent alterations. The frequency of most common alterations per histology type. Total number of distinct alterations *N* = 546. The graph displays unique alterations that occur in ≥3% of patients with the specified histology.

**Figure 5 cancers-11-00011-f005:**
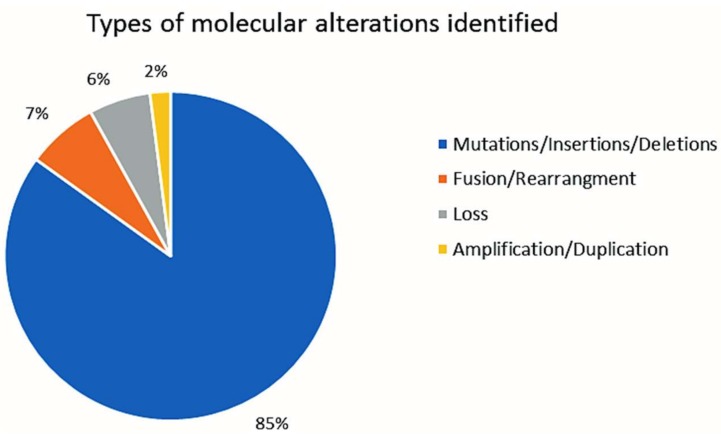
Types of molecular alterations identified.

**Figure 6 cancers-11-00011-f006:**
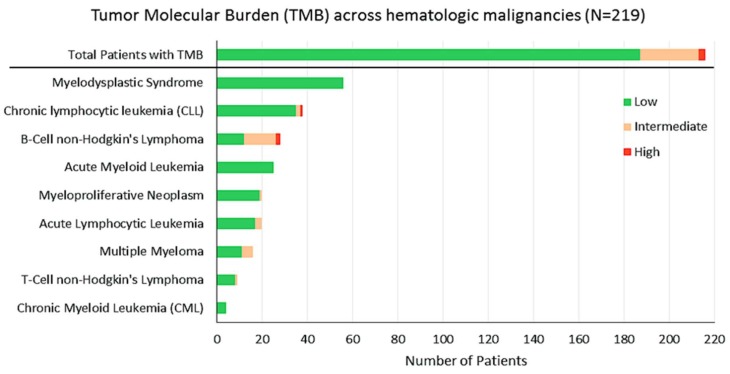
Tumor Mutation Burden (TMB) across hematologic malignancies. Note: *N* = 219 total number of patients analyzed for TMB. The majority of patients (84%) had low TMB (≤1–5 alterations/MB), 12% were found to have intermediate TMB (>5 to ≤19 alterations/MB), and only 2% of the patients harbored high TMB (≥20 alterations/MB).

**Table 1 cancers-11-00011-t001:** Patient characteristics (*N* = 227 patients with adequate tissue for comprehensive genomic profiling (CGP)).

Patient Demographics and Baseline Characteristics	
**Gender:** Men (%); Women (%)	133 (59%); 94 (41%)
**Age:** Median age (range)	59 years (17–88)
**Ethnicity:** Caucasian (%)	163 (71.8%)
Asian (%)	18 (8%)
Hispanic (%)	18 (8%)
African American (%)	8 (3.5%)
Other (%)	20 (8.8%)
**Histologies:**	
**Myeloid Disorders**	***N* = 124 (54.6%)**
Myelodysplastic syndrome (MDS)	52 (22.9%)
Acute myeloid leukemia (AML)	25 (11%)
Myeloproliferative neoplasia (MPN)	21 (9.2%)
Multiple Myeloma (MM)	17 (7.5%)
Chronic myeloid leukemia (CML)	4 (1.8%)
Other myeloid disorders	5 (2.2%)
**Lymphoid disorders**	***N* = 103 (45.4%)**
Chronic lymphocytic leukemia (CLL)	39 (17.2%)
Acute lymphocytic leukemia (ALL)	20 (8.8%)
Diffuse large B-cell lymphoma (DLBCL)	18 (7.9%)
Follicular lymphoma (FL)	6 (2.6%)
Marginal zone lymphoma (MZL)	4 (1.8%)
Anaplastic large cell lymphoma (ALCL)	2 (0.9%)
Castleman disease	2 (0.9%)
Other lymphoid disorders	12 (5.3%)
**Summary of alterations**	
Number of patients with alterations	197 (87%)
**Number of patients with potentially actionable alterations**	170 (75%)
Median number of alterations/patient (range)	3 (0–14)
Median number of potentially actionable alterations/patient (range)	1 (0–7)
Total alterations	698
Number of distinct alterations	546
Number of distinct potentially actionable alterations	256

**Table 2 cancers-11-00011-t002:** Genomic alterations and examples of potential targeted (either on- or off- label) therapeutic.

Gene Alteration	Gene Function	Examples of Potential on/off-Label Therapy	Examples of Potential Experimental Therapy/Clinical Trial *	Ref.
*ABL1/2*	ABL (Abelson tyrosine-protein) kinase regulates cell survival and division/differentiation	Imatinib, Dasatinib, Nilotinib, Bosutinib, Ponatinib		
*APC*	APC (adenomatous polyposis coli) is a tumor suppressor, regulating cell division/adhesion, controls Wnt signaling pathway	Sulindac (Tankyrase inh)		[13]
ARID1A	ARID1A (AT-rich interactive domain-containing protein 1A) regulates transcription	Dasatinib, EZH2 inh.	Talazoparib Tosylate NCT02286687 **	
ASXL1	ASXL1 (additional sex combs-like1) regulates transcription and ubiquitin-proteasome protein degradation via BAP pathway.	Cabozantinib		[14]
*ATM*	ATM (ataxia telangiectasia mutated) regulates DNA damage response via the PI3K-like protein kinase pathway	Olaparib		[15]
*BCL2*	BCL2 (B-cell lymphoma 2) regulates apoptosis	Venetoclax		[16]
*BRAF*	BRAF regulates cell growth via MAPK (RAF-MEK-ERK) signaling cascade	Dabrafenib, Regorafenib, Trametinib, Vemurafenib, Cobimetinib		[17]
*BRCA2*	BRCA2 (breast cancer 1/2) regulates DNA double-strand break repair	Olaparib, Niraparib, Rucaparib		
BRIP1	BRIP1 (BRCA1-interacting protein 1) functions in DNA repair	Olaparib		
*BTK*	BTK (Bruton’s tyrosine kinase) regulates B-cell receptor signaling and B-cell development	Ibrutinib, Acalabrutinib		[18]
*CCND2*	CCND1/3 (cyclin D1/3) regulates cell cycle via CDK4/6	Palbociclib		[19]
*CD274*	CD274 (cluster of differentiation 274) encodes immune inhibitory receptor B7-H1, also known as programmed cell death ligand-1 (PD-L1)	Atezolizumab, Avelumab, Durvalumab, Nivolumab, Pembrolizumab		
*CD79B*	CD79A/B (cluster of differentiation 79) complexes with B-cell receptor, mediates downstream signaling to the NF-kB, PI3K, MAPK and NF-AT pathways	Ibrutinib	Polatuzumab vedotin	[20]
*CDK4*	CDK4 (cyclin-dependent kinase 4) regulates cell cycle	Palbociclib, Ribociclib		[19]
*CDKN2A/B*	CDKN2A (cyclin dependent kinase inhibitor encodes tumor suppressors and regulates cell cycle; loss results in increased CDK4/6	Palbociclib, Ribociclib		[19]
*CSF1R*	CSF1 (colony stimulating factor 1) regulates differentiation and survival		Chiauranib NCT03074825 **	
*CXCR4*	CXCR4 (C-X-C chemokine receptor type 4) regulates hematopoiesis and CD20 expression	Plerixafor	BMS-936564 NCT01120457 **	[21]
*DNMT3A*	DNMT3A (DNA methyltransferase 3A) regulates gene expression	Azacitidine, Decitabine		[22]
*EP300*	Histone acetyltransferase p300 regulates transcription via chromatin remodeling		Mocetinostat NCT02282358 **	
*ERBB4*	Member of the EGFR (epidermal growth factor receptor) regulates proliferation	Trastuzumab, Pertuzumab Afatinib, Erlotinib, Lapatinib		
*EZH2*	EZH2 (enhancer of zeste-homolog 2) regulates DNA methylation and transcription repression		Tazemetostat (NCT02601950) **	[23]
*FGFR3*	FGFR3 (fibroblast growth factor receptor 3) promotes cell cycle via activation of RAS/MAPK/AKT pathway	Lenvatinib, Pazopanib, Ponatinib, Regorafenib		
*FLT3*	FLT3 (FMS-like tyrosine kinase 3) activates signaling of Akt1, RAS, ERK, and mTOR.	Midostaurin, Gilteritinib Quizartinib		[9,24]
*FLT4*	FLT4 (FMS like tyrosine kinase 4), also known as VEGFR-3 (vascular endothelial growth factor receptor 3)	Sorafenib, Sunitinib, Pazopanib, Axitinib, Vandetanib,		[25]
*GNAS*	GNAS (Guanine nucleotide binding protein, α stimulating) regulates adenylate cyclase via MAPK	Trametinib		
*IDH1*	IDH1 (isocitrate dehydrogenases 1)	Azacitidine, Decitabine		[26]
*IDH/2*	IDH2 (isocitrate dehydrogenases 2) regulates citric acid (Krebs) cycle and cell metabolism	Enasidenib		[27]
*IGF1R*	IGF1R (insulin-like growth factor-1 receptor) mediates anti-apoptotic signals		Ganitumab NCT00562380 **	
*JAK1*	JAK1 (Janus kinase 1) in involved in signal regulation	Tofacitinib	Fedratinib	[28]
*JAK2*	JAK2 (Janus kinase 2) is involved in signal regulation	Ruxolitinib		[29]
*KIT*	KIT (also known as c-Kit or CD117), activates PI3K/Akt and RAS/MAPK signaling pathway	Imatinib, Midastaurin		
*KRAS*	KRAS (Kirsten rat sarcoma) regulates signal transduction via MAPK pathway	Cetuximab, Trametinib, Panitumumab, Regorafenib		[30]
*MAP2K1*	MAP2K1 (mitogen-activated protein kinase 1 (MKK1 or MEK1) mediates RAS/RAF/MAPK pathway	Cobimetinib, Selumetinib, Trametinib		
*MAP3K14*	MAP3K14 (mitogen-activated protein kinase 14) also known as NF-kappa-B-inducing kinase	Trametinib		
MLL	MLL (mixed lineage leukemia) encodes a histone methyltransferase		EPZ-5676 NCT02141828 **	
*MSH2*	MSH2 (MutS homolog2) is a tumor suppressor encodes DNA mismatch repair (MMR) protein 2	Atezolizumab, Nivolumab Pembrolizumab		
*MSH6*	MSH6 (MutS homolog 6) encodes DNA mismatch repair (MMR) protein 6 involved in DNA repair	Atezolizumab, Nivolumab Pembrolizumab		
MYC	MYC regulates cell cycle progression, apoptosis, proliferation		BET inhibitors NCT02431260 **	
MYD88	MYD88 (myeloid differentiation primary response gene 88) activates transcription factor NFkB	Ibrutinib, acalabrutinib (IRAK1 inh)	zanubrutinib	[7]
NF1/2	NF1 (neurofibromin 1/2) a GTPase-activating negative regulator of the RAS signaling pathway	Everolimus, Temsirolimus, Trametinib		
NRAS	NRAS (neuroblastoma RAS) mediates signal transduction via RAF/MEK/ERK and PI3K	Trametinib, Panitumumab		[31]
PALB2	PALB2 (partner and localizer of BRCA2)	Olaparib		[32]
PDCD1LG2	Programmed cell death 1 ligand 2 (also known as CD273) essential for T-cell proliferation	Atezolizumab, Avelumab, Durvalumab, Nivolumab, Pembrolizumab		
PIK3CA	PIK3CA (phosphatidylinositol 3-kinase (PI3K), which regulates the PI3K/AKT/MTOR axis	Everolimus, Temsirolimus, Copanlisib, Duvalisib, Idelalisib	Taselisib NCT02465060 **	
PIK3R1	PIK3R1 (PI3K regulatory subunit alpha)	Copanlisib	NCT02369016 **	
PTCH1	PTCH1 (Protein patched homolog 1) is a receptor for Sonic hedgehog (Shh) for gene transcription	Vismodegib, Sonidegib		[33]
PTEN	PTEN (phosphatase and tensin homolog) is a tumor suppressor, functions via PI3K/AKT/mTOR pathway	Everolimus, Temsirolimus		[34]
PTPN11	PTPN11 (Tyrosine-protein phosphatase non-receptor type 11) activates PI3K, MEK axis	Trametinib,		[35]
RET	RET (rearranged during transfection) is a proto-oncogene	Cabozantinib, Sorafenib, Vandetanib, Lenvatinib		[36]
RUNX1	RUNX1 (Runt-related transcription factor, also known as acute myeloid leukemia 1 protein (AML1), core-binding factor subunit alpha 2 (CBFA2) is a tumor suppressor		Mocetinostat (MGCD0103) or Sorafenib NCT00217646 **	
STAT3	STAT3 (signal transducer and activator of transcription 3) encodes a transcription factor		AZD9150 (NCT01839604) **	
STK11	STK11 (serine/threonine kinase 11) functions as a tumor suppressor gene	Dasatinib, Bosutinib, Everolimus, Temsirolimus		
TET2	TET2 (Tet methylcytosine dioxygenase 2) regulates DNA demethylation	Azacitidine, Decitabine		[37]
TP53	TP53 (Tumor protein p53) is a tumor suppressor; loss leads to overexpression of VEGF levels	Bevacizumab, Pazopanib	Wee-1 inh, MDM inh, PRIMA-1MET inhibitors.	[38]
VHL	VHL (von Hippel-Lindau) is a tumor suppressor activates the HIF/VEGF pathway	Axitinib, Bevacizumab, Everolimus, Pazopanib, Sorafenib, Sunitinib, Temsirolimus, Vandetanib,		
XPO1	XPO1 (exportin-1) regulates nuclear export of tumor suppressor genes		Selinexor NCT02227251	

* Experimental drugs in clinical trials are generally only mentioned if there are no FDA-approved drugs that impact that target. ** Numbers refer to clinicaltrials.gov identifier (https://clinicaltrials.gov/ct2/show).

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
