# Peer review of "Comprehensive Genomic Profiling Reveals Diverse but Actionable Molecular Portfolios across Hematologic Malignancies: Implications for Next Generation Clinical Trials"

_cancers, 2018, doi:10.3390/cancers11010011_

Round 1
Reviewer 1 Report
In their paper Galanina et al., are describing a cohort of 227 patients with different haematological cancers that was used to identify molecular targets via NGS and their potential actionability in target therapies. The paper is well written and provides interesting insights into the genomic landscape of these patients and an interesting rationale in using NGS as an informative tool for blood cancers similar to what happens for solid malignancies.
Comments:
1) The title of the paper is a bit long and confusing. The authors should consider to rephrase this.
2) The cohort of patients is well described and summarised in table 1.However, the authors should clarify details of the time point for the samples in use (eg: at diagnosis) “The most common tissue source for NGS analysis was peripheral blood (N = 99 93 patients [44%]) or bone marrow (N = 86 [38%]), followed by lymph node (N = 17 [7%]) and other (N 94 = 25 [11%])”.
3) In their abstract, under the results section, the authors describe a cohort of 227 (96.5%) patients suitable for their studies. They then split these according to their diagnosis but if adding the percentage provided in brackets these do not match the total of 96.5%. This should be clarified. The same is also from line 90 in the main text.
4) The percentage also do not match in Table 1 under the Ethnicity section (total 101%).
5) Colours in figure 4 are very bright (on PDF) and should be reconsidered specially the yellow for AML that is barley readable.
Author Response
Comments and Suggestions for Authors
In their paper Galanina et al., are describing a cohort of 227 patients with different haematological cancers that was used to identify molecular targets via NGS and their potential actionability in target therapies. The paper is well written and provides interesting insights into the genomic landscape of these patients and an interesting rationale in using NGS as an informative tool for blood cancers similar to what happens for solid malignancies.
Comments:
1) The title of the paper is a bit long and confusing. The authors should consider to rephrase this.
RESPONSE: The title of the paper:
“Comprehensive genomic profiling reveals complex and distinct but actionable molecular portfolios across hematologic malignancies: Implications for next generation clinical trials” has been modified to: “Comprehensive genomic profiling reveals diverse but actionable molecular portfolios across hematologic malignancies: Implications for next generation clinical trials
2) The cohort of patients is well described and summarised in table 1.However, the authors should clarify details of the time point for the samples in use (eg: at diagnosis) “The most common tissue source for NGS analysis was peripheral blood (N = 99 patients [44%]) or bone marrow (N = 86 [38%]), followed by lymph node (N = 17 [7%]) and other (N 94 = 25 [11%])”.
RESPONSE: Most of the tissue sequencing was done at diagnosis (some were done at the time of relapse).
3) In their abstract, under the results section, the authors describe a cohort of 227 (96.5%) patients suitable for their studies. They then split these according to their diagnosis but if adding the percentage provided in brackets these do not match the total of 96.5%. This should be clarified. The same is also from line 90 in the main text.
RESPONSE. The total number of patients examined for the study was 235. However, 8/235 (3%) had inadequate tissue for analysis and were therefore excluded; the remainder 227/235 (96.5%) did have sufficient tissue. Only patients with adequate tissue were used for all further analyses (i.e. 227 is the denominator). Therefore, all histologies and tissue source data add up to 227 (which is the entire analyzable cohort).
4) The percentage also do not match in Table 1 under the Ethnicity section (total 101%).
RESPONSE: The percentages were rounded off to the nearest whole number. To fix this issue we have made adjustments to round off to the nearest decimal point.
5) Colours in figure 4 are very bright (on PDF) and should be reconsidered specially the yellow for AML that is barley readable.
RESPONSE: Thank you for pointing this out. I will try to adjust saturation/intensity for the final (printed) version of the manuscript.

Reviewer 2 Report
In this manuscript, authors analyzed alterations in 406 genes in a cohort of 235 patients with different hematologic malignancies by integrated NGS profiling, and found that most patients harbored one or more genomic alterations that could be targeted by available drugs or immunotherapies. The authors conclude that therapies matched to genomic portfolios could be useful in treating hematologic malignancies, and clinical trials to test this concept are warranted.
This manuscript is of clinical interest and well written. The data is well presented and convincing.
Minor comments:
1. Authors should describe “Consolidated Standards of Reporting Trials” in the introduction section. The term came out in sudden in Figure 1 legend.
2. Authors should also run analysis based on age, and compared with age-matched healthy controls to see whether the genetic alterations observed in patient cohort are cancer-specific.
3. It may be better if authors could show some ex-vivo data to demonstrate the effects of targeted therapies based on genetic alterations. For example, use ibrutinib and trametinib to treat samples with MYD88 and NRAS mutations.
Author Response
In this manuscript, authors analyzed alterations in 406 genes in a cohort of 235 patients with different hematologic malignancies by integrated NGS profiling, and found that most patients harbored one or more genomic alterations that could be targeted by available drugs or immunotherapies. The authors conclude that therapies matched to genomic portfolios could be useful in treating hematologic malignancies, and clinical trials to test this concept are warranted.
This manuscript is of clinical interest and well written. The data is well presented and convincing.
Minor comments:
1. Authors should describe “Consolidated Standards of Reporting Trials” in the introduction section. The term came out in sudden in Figure 1 legend.
RESPONSE: This is a standard nomenclature and is described in the figure legend.
2. Authors should also run analysis based on age, and compared with age-matched healthy controls to see whether the genetic alterations observed in patient cohort are cancer-specific.
RESPONSE: This is an excellent point. Thank you. Unfortunately, we do not have a database of healthy volunteers yet. We are in the process of getting that set up. In the future, this willt be a very important concept to look further at clonal hematopoiesis of indeterminate potential (CHIP) to determine wich alterations are oncogenic and cancer specific.
3. It may be better if authors could show some ex-vivo data to demonstrate the effects of targeted therapies based on genetic alterations. For example, use ibrutinib and trametinib to treat samples with MYD88 and NRAS mutations.
RESPONSE: Thank you very much for the insightful comment. Yes, we are collecting outcomes data now in an effort to determine the efficacy of molecularly targeted therapy. This is in process and will be reported in a separate paper.

Reviewer 3 Report
This is a concisely written review on current aspects of the rapidly evolving field of molecular diagnostics with relevance for clinical decision making. The review may turn out to be useful for readers.
No obvious misinformation is detectable to this reviewer. The authors may want to update the recent FDA approvals in the field.
Author Response
This is a concisely written review on current aspects of the rapidly evolving field of molecular diagnostics with relevance for clinical decision making. The review may turn out to be useful for readers.
No obvious misinformation is detectable to this reviewer. The authors may want to update the recent FDA approvals in the field.
RESPONSE: We thank the reviewer for positive feedback. The field of targeted therapy is moving very rapidly, all the relevant updates have been done.
